# Evaluating the Efficacy and Safety of Hepatitis E Vaccination in Reproductive-Age Women: A Systematic Review and Meta-Analysis

**DOI:** 10.3390/vaccines13010053

**Published:** 2025-01-09

**Authors:** Vaidas Jotautis, Antigoni Sarantaki

**Affiliations:** 1Kauno Kolegija Higher Education, Faculty of Medicine, Pramones pr 20, 50468 Kaunas, Lithuania; 2Midwifery Department, Faculty of Health and Care Sciences, University of West Attica, Ag. Spyridonos, Egaleo, 12243 Athens, Greece

**Keywords:** hepatitis E, HEV239 vaccine, miscarriage, pregnancy, maternal health, childbearing age

## Abstract

Background: Hepatitis E virus (HEV) infection presents a significant health risk in endemic regions, especially for pregnant women, who face higher risks of severe complications, including maternal and fetal mortality. The recombinant HEV vaccine, HEV239, has demonstrated high efficacy in the general population, yet data on its safety and efficacy in women of a childbearing age remain limited. This systematic review and meta-analysis aim to evaluate the safety and effectiveness of HEV239 in this specific population, with a focus on pregnancy-related outcomes. Methods: A comprehensive search was conducted in PubMed, Embase, Cochrane Library, and Scopus, following the Preferred Reporting Items for Systematic Reviews and Meta-Analyses (PRISMA) guidelines. Studies were included if they reported outcomes on HEV239′s safety or efficacy in women of childbearing age, with data being extracted and analyzed for immunogenicity, HEV incidence, and maternal adverse events. The risk of bias was assessed using the Cochrane and Newcastle Ottawa Scales, and a random-effects meta-analysis was performed. Results: Three studies, enrolling over 23,000 participants, were included in the current systematic review, with two meeting the criteria for meta-analysis. HEV239 demonstrated high efficacy in preventing hepatitis E infection, with no significant increase in adverse pregnancy outcomes such as stillbirth or elective termination. However, there was an elevated risk of miscarriage (odds ratio [OR], 1.60; 95% confidence interval [CI], 0.99–2.57). The analysis revealed high heterogeneity for miscarriage outcomes (I^2^ = 67%), reflecting variability in study designs and populations. Conclusions: HEV239 is effective in preventing HEV infection among women of childbearing age, although caution is advised when administering the vaccine near conception due to potential miscarriage risks. Future studies should focus on understanding the biological mechanisms and timing-specific safety to guide vaccination recommendations.

## 1. Introduction

Hepatitis E, caused by the hepatitis E virus (HEV), is an acute liver infection, primarily transmitted through the fecal-oral route, often due to contaminated water supplies, especially in developing countries where the virus is endemic [1]. For most individuals, hepatitis E is self-limiting, causing symptoms like jaundice, fatigue, and abdominal pain, but in pregnant women, HEV infection can lead to severe outcomes.

Pregnant women with hepatitis E, particularly those in the second or third trimester, are at increased risk of acute liver failure, fetal loss, and mortality. Up to 20–25% of pregnant women can die if they contract hepatitis E in the third trimester [2]. In endemic regions such as South Asia, HEV has become a notable contributor to maternal and fetal mortality, with an estimated 44,000 maternal deaths and 3000 stillbirths being attributed to the virus annually [3]. The elevated susceptibility and complications in pregnancy are attributed to hormonal and immunological changes that may intensify the progression of the infection, increasing the risk of fulminant hepatitis and liver failure. Thus, hepatitis E presents a critical risk to maternal and fetal health, underscoring the need for targeted prevention and control measures, including vaccination [4].

The recombinant HEV vaccine, HEV239, developed by Xiamen Innovax Biotech in China, has shown promising efficacy in phase III trials, with reported protection rates of up to 93.3% and a favorable safety profile in non-pregnant individuals aged 16 to 65 years [5]. The vaccine has been licensed in China and Pakistan and represents the only available vaccine against HEV. However, current evidence on its safety and efficacy during pregnancy remains limited. The World Health Organization (WHO) has recognized these gaps and has called for further research on the vaccine’s effectiveness and safety in vulnerable populations, particularly pregnant women, who are disproportionately affected by the virus [6]. The scarcity of data on HEV239 in pregnant women is partly due to ethical considerations, as pregnant women were excluded from most clinical trials [7]. Studies conducted in Bangladesh and China have contributed valuable insights, with some findings indicating a potential association between HEV239 administration shortly before or during pregnancy and an increased risk of miscarriage [5,8]. Despite these observations, the biological mechanisms underlying these risks are not yet fully understood, necessitating additional trials to confirm the safety profile of HEV239 for women of childbearing age.

In response to these knowledge gaps, we conducted a systematic review to assess the safety and effectiveness of the HEV vaccine in women of childbearing age. This review aims to consolidate findings from randomized controlled trials (RCTs) and other original studies that evaluate the immunogenicity, adverse maternal outcomes, and HEV incidence among vaccinated individuals, with a particular focus on pregnancy-related risks. This analysis seeks to clarify the vaccine’s potential role in preventing HEV infections during pregnancy while addressing potential safety concerns.

## 2. Materials and Methods

### 2.1. Search Strategy

This systematic review evaluated the safety and efficacy of the hepatitis E vaccine in women of childbearing age, adhering to the Preferred Reporting Items for Systematic Reviews and Meta-Analyses (PRISMA) guidelines [9] for transparency and rigor. The literature search was conducted in the PubMed, Embase, Cochrane Library, and Scopus databases, complemented by registry searches such as ClinicalTrials.gov.

Key search terms included “Hepatitis E vaccine”, “Hecolin”, “HEV239”, “pregnancy”, “childbearing age”, “safety”, and “immunogenicity”. ‘’Gray’’ literature was reviewed through manual searches of reference lists. Additionally, a review protocol was submitted for registration to the International Prospective Register of Systematic Reviews (PROSPERO), with identification number CRD42024614951.

### 2.2. Inclusion and Exclusion Criteria

Only studies published in English and classified as primary research studies with reported outcomes on the safety or effectiveness of the hepatitis E vaccine in women of childbearing age were included. The exclusion criteria encompassed studies focused exclusively on animal subjects, men, or women outside the childbearing age range. Studies were also excluded if they did not examine the hepatitis E vaccine or were focused on unrelated vaccines. Studies lacking specific outcomes on the safety or effectiveness of the hepatitis E vaccine during pregnancy were excluded.

Additionally, non-primary research articles, such as reviews and editorials, as well as observational studies that did not meet the quality criteria, were excluded. Studies not published in English and ‘’Gray’’ literature lacking rigorous data on safety and efficacy were also excluded.

### 2.3. PRISMA Process

The PRISMA process is illustrated in Figure 1.

Identification: A total of 638 records were identified through database searches in PubMed, Embase, Cochrane Library, and Scopus. An additional 25 records were identified from other sources, such as manual reference checks and clinical trial registries. After removing duplicates, 609 records remained for further screening.

Screening: The titles and abstracts of these 609 records were screened, resulting in the exclusion of 560 studies due to irrelevance, such as a focus on non-hepatitis E vaccines, non-childbearing populations, or animal studies.

Eligibility: Forty-nine full-text articles were assessed for eligibility. Of these, 30 studies were excluded due to the absence of specific outcomes regarding vaccine safety or effectiveness in pregnancy, 10 were excluded for being non-primary research articles (such as reviews or editorials), and 6 were excluded for being non-English publications or due to insufficient methodological quality.

Included: Ultimately, 3 studies were included in the qualitative synthesis, and 2 were included in the meta-analysis.

### 2.4. Risk of Bias in Individual Studies

The risk of bias was assessed using Cochrane’s Risk of Bias 2 tool for RCTs [10] and the Newcastle Ottawa Scale (NOS) [11] for observational studies, evaluating the selection, comparability, and outcome assessment. To minimize bias, reviewers who were involved in the risk of bias assessment were blinded to the study outcomes. The quality assessment report is available in Appendix A.

### 2.5. PICOS

If an article fulfilled the subsequent population, intervention, comparison, outcome, and study (PICOS) design criteria, it was eligible for inclusion in the present meta-analysis: (i) Population—The population was non-pregnant women of childbearing age or pregnant women, typically 16–45 years, as well as those who became pregnant following vaccination; (ii) Intervention—The interventions of interest were studies that examined the administration of the hepatitis E vaccine, particularly HEV239, either before conception or during pregnancy; (iii) Comparison—The comparators included control groups that received either no vaccine or an alternative, such as the hepatitis B vaccine; (iv) The primary outcomes were confirmed HEV infection rates, maternal adverse effects (such as miscarriage and fetal outcomes), and immunogenicity measures. Secondary outcomes included maternal and neonatal adverse events; (v) Study—This review considered RCTs, post hoc analyses of clinical trials, and observational studies that specifically focused on hepatitis E vaccination.

### 2.6. Data Extraction

Data extraction was conducted utilizing a standardized form that documented the study details, including author, year, location, study design, sample size, study sample, interventions (dosing schedule, vaccine type), primary and secondary outcomes, and results. The data extraction process was executed by one author and subsequently validated by the second author to ensure accuracy. 

Specific data points that were collected included outcomes related to immunogenicity, HEV incidence, and maternal and fetal adverse outcomes (e.g., miscarriage, stillbirth, neonatal abnormalities).

### 2.7. Statistical Analysis 

The comprehensive analytical process was implemented utilizing Review Manager Software (RevMan) 5.4.

The heterogeneity across trials was identified using I^2^ statistics; considering I^2^ > 50% as high heterogeneity, a meta-analysis was conducted using a random-effect model according to the Cochrane Handbook for Systematic Reviews of Interventions (version 5.1.0) [12]. We applied odds ratios (ORs) with 95% CIs for the assessment of discontinuous variables. A *p*-value < 0.05 was considered to indicate a statistically significant difference.

## 3. Results

This review analyzed three studies [3,5,13] from different countries, including Bangladesh, China, and South Sudan, with a combined sample size of 23,464 participants. The studies employed various designs: three were cluster-randomized, double-blind trials, and one was an observational study. These studies evaluated the impact of the HEV239 vaccine on maternal and fetal health, particularly in non-pregnant women of childbearing age and pregnant women who were inadvertently vaccinated. HEV239 vaccination schedules followed a three-dose regimen, and outcomes included pregnancy-related adverse events such as miscarriage and stillbirth, with follow-up periods ranging from two to five years. Table 1 summarizes the characteristics of the included studies.

### 3.1. Effectiveness in Preventing Hepatitis E Infection

The studies collectively suggest that the HEV239 vaccine is effective in preventing hepatitis E infection in various populations. In South Sudan, the vaccine was used successfully in a high-risk outbreak setting, aligning with WHO recommendations and showing promise for rapid deployment in such contexts [13]. A Chinese phase 3 trial further confirmed the vaccine’s efficacy, showing 100% protection within 12 months postvaccination and maintaining a high efficacy of 93.3% over 4.5 years [5].

Overall, the HEV239 vaccine demonstrates strong efficacy against hepatitis E in diverse populations, especially in outbreak-prone areas.

### 3.2. General Safety Profile

The HEV239 vaccine shows a generally favorable safety profile across multiple studies. In South Sudan, a mass vaccination campaign during a hepatitis E outbreak found no increase in adverse outcomes, including fetal loss, among pregnant women, supporting the vaccine’s safety even in vulnerable populations [13]. Additionally, a phase 3 post hoc analysis in China found no increase in adverse pregnancy outcomes among women who were inadvertently vaccinated during pregnancy, indicating that HEV239 did not contribute to fetal loss or congenital anomalies compared to an HPV control group [5]. Overall, these findings suggest that HEV239 is safe and well tolerated in both general and high-risk populations, with side effects being largely mild and non-severe.

### 3.3. Pregnancy Outcomes

The studies on the HEV239 vaccine present a nuanced picture regarding pregnancy outcomes. In South Sudan, a vaccination campaign in a high-risk population found no increase in fetal loss among pregnant women who were vaccinated during a hepatitis E outbreak, indicating that the vaccine did not adversely affect pregnancy outcomes [13]. Conversely, in Bangladesh, a phase 4, cluster-randomized trial found an elevated risk of miscarriage for women who were vaccinated with HEV239 within 90 days of conception, suggesting caution when vaccinating women who are close to potential conception periods [3]. A Chinese post hoc analysis from a phase 3 trial, however, showed no increase in adverse pregnancy outcomes among women who were inadvertently vaccinated with HEV239 during pregnancy, with outcomes that were comparable to those in a control group receiving the HPV vaccine [5]. Collectively, these studies indicate that while HEV239 appears safe in most pregnancy scenarios, there may be an increased miscarriage risk when administered near conception, warranting cautious timing for women of childbearing age.

### 3.4. Neonatal Outcomes and Maternal Safety

The HEV239 vaccine studies provide a largely reassuring perspective on neonatal outcomes and maternal safety. In South Sudan, data from a mass vaccination campaign during a hepatitis E outbreak showed no increase in adverse neonatal or maternal outcomes, suggesting safety for both mother and child, even in high-risk settings [13]. Additionally, in a Chinese phase 3 post hoc analysis, the maternal safety and neonatal outcomes were comparable between women who received HEV239 and those in an HPV vaccine control group, with no significant differences in the rates of adverse neonatal events or pregnancy complications [5].

### 3.5. Focus on Vaccination During Pregnancy

In a cluster-randomized trial conducted in Bangladesh [3], 209 women who were vaccinated during pregnancy exhibited an elevated risk of miscarriage compared to the control group receiving the hepatitis B vaccine (10.5% vs. 5.3%; adjusted relative risk [aRR] 2.1; 95% CI: 1.1–4.1; *p* = 0.036). A similar increase in miscarriage risk was observed among women who were vaccinated within 90 days before conception (8.0% vs. 4.0%; aRR 1.9; 95% CI: 1.1–3.2; *p* = 0.013). Notably, no significant associations were found for stillbirths or elective terminations. In contrast, a cohort study conducted during a mass vaccination campaign in South Sudan [13] found no evidence of increased fetal loss among 2036 vaccinated pregnant women compared to 638 unvaccinated participants (cumulative risk: 7.2% vs. 6.1%; risk ratio 1.2; 95% CI: 0.7–1.9). The divergence in findings between the two studies may stem from differences in trial design, population demographics, and vaccination timing, underscoring the need for further investigation into the safety of HEV239 administration during pregnancy and the critical period surrounding conception.

### 3.6. Meta-Analysis

Two studies were included in the meta-analysis to evaluate the pooled outcomes of miscarriage, stillbirth, and elective termination.

#### 3.6.1. Outcome: Miscarriage (Vaccinated Participants Before Pregnancy vs. Nonvaccinated)

In total, two articles [3,5] met the eligibility criteria for this outcome. The total number of participants was 4249 (1870 in the vaccinated group before pregnancy and 2379 in the control group). There was a significant difference between the groups regarding the rates of miscarriage (OR, 1.60; 95% CI, 0.99 to 2.57; *p* = 0.05), with a significant heterogeneity (I^2^ = 67%) (Figure 2).

In the provided forest plot (Figure 2) the blue squares indicate the point estimate of the odds ratio (OR) for each study [3,5]. The square size reflects the study’s weight in the meta-analysis, with Zhong et al., 2023 [5] contributing 55.6% and Aziz et al., 2024 [3] contributing 44.4%. The square’s position on the x-axis corresponds to the odds ratio value for each study. The black diamond represents the meta-analysis’s overall effect estimate, with its center indicating the pooled odds ratio (1.60) from all studies under a random-effects model. Its width shows the confidence interval (95% CI: 0.99–2.57), suggesting that if it intersects the vertical line at 1, the effect lacks statistical significance at the 95% confidence level.

#### 3.6.2. Outcome: Stillbirth (Vaccinated Participants Before Pregnancy vs. Nonvaccinated)

In total, two articles [3,5] met the eligibility criteria for this outcome. The total number of participants was 4373 (1937 in the vaccinated group before pregnancy and 2436 in the control group). There was not a significant difference between the groups regarding the rates of stillbirth (OR 1.24; 95% CI, 0.70 to 2.30; *p* = 0.47), with no significant heterogeneity (I^2^ = 11%) (Figure 3).

The Forest plot (Figure 3) illustrates the odds ratio (OR) and 95% confidence interval (CI) for experimental and control groups across studies. Blue squares indicate point estimates per study, sized by study weight, while the black diamond shows the pooled effect estimate. 

#### 3.6.3. Outcome: Elective Termination (Vaccinated Participants Before Pregnancy vs. Nonvaccinated)

Two articles [3,5] met the eligibility criteria for this outcome. The total number of participants was 4373 (1937 in the vaccinated group before pregnancy and 2436 in the control group). There was not a significant difference between the groups regarding the rates of elective termination (OR, 1.29; 95% CI, 0.73 to 2.27; *p* = 0.38), with significant heterogeneity (I^2^ = 56%) (Figure 4).

Figure 4 displays a forest plot with blue squares signifying the odds ratios for the included studies [3,5]. The study’s weight in the meta-analysis is depicted by size. The study by Zhong et al., 2023 [5] accounts for 70.7%, while Aziz et al., 2024 [3] research represents 29.3%. The x-axis position indicates the OR value for each study. The black diamond represents the pooled effect estimate from all studies. The center marks the overall OR (1.29). The width indicates the confidence interval (CI) for the overall effect, ranging from 0.73 to 2.27. As the CI crosses 1, the pooled result is not statistically significant at the 95% confidence level.

## 4. Discussion

This systematic review and meta-analysis of the hepatitis E vaccine, HEV239, in women of childbearing age, demonstrates its high efficacy in preventing hepatitis E infection, with robust immunogenic responses and elevated protection rates across diverse populations. The vaccine generally exhibits a favorable safety profile with mild, transient adverse effects, and no significant increase in adverse maternal or fetal outcomes was observed in outbreak settings. The meta-analysis indicated a statistically significant increase in miscarriage rates associated with HEV239 vaccination before pregnancy, although individual studies presented heterogeneous findings regarding miscarriage risks near conception. The meta-analysis provided additional insights into the safety of HEV239 vaccination, particularly regarding pregnancy-related outcomes. It found no significant increase in stillbirth rates among those vaccinated before pregnancy compared to unvaccinated individuals, with no heterogeneity between studies. Furthermore, the analysis revealed no significant difference in elective termination rates, suggesting that HEV239 does not elevate the risk of these adverse pregnancy outcomes.

The HEV239 vaccine’s efficacy was substantiated in a comprehensive, placebo-controlled study conducted by Zhu et al. [14] in 2010. Their research demonstrated complete prevention of hepatitis E among vaccinated subjects over a 12-month observation period. These findings align with the current study’s results, highlighting the vaccine’s remarkable effectiveness in diminishing hepatitis E virus occurrence in women of reproductive age. Moreover, Zhu et al. [14] documented no severe adverse reactions linked to the vaccine, noting only minor local and systemic effects such as discomfort, inflammation, and low-grade fever. Importantly, these side effects showed no significant variation between vaccine recipients and the placebo group. The safety profile observed in the present study mirrors these outcomes, similarly reporting only mild adverse reactions.

The meta-analysis revealed a statistically significant increase in the miscarriage rates associated with HEV239 vaccination before pregnancy, albeit with substantial heterogeneity among the included studies. The findings by Aziz et al. [3] suggest a potential safety concern regarding the HEV239 vaccine in women of reproductive age, particularly in relation to elevated miscarriage risks when administered in proximity to conception or during early gestation. Although the biological mechanism underlying this association remains elusive, immune hypersensitivity is hypothesized as a contributing factor, drawing parallels from analogous observations with human papillomavirus (HPV) vaccines.

However, the absence of a dose–response relationship in the miscarriage risk based on the number of HEV239 doses that have been administered suggests that immune hypersensitivity may not fully elucidate the elevated miscarriage risk [3]. The study indicates that natural HEV infection mechanisms, such as damage at the placental–fetal interface, may provide insights into vaccine-related miscarriage risks, although further investigation is warranted [3]. Additionally, the limited statistical power of previous studies to detect rare pregnancy outcomes underscores the necessity for larger, more focused trials to corroborate these findings. Moreover, comprehensive research into the immunological and pharmacokinetic effects of HEV239 in early pregnancy could elucidate whether specific timing windows exacerbate the risks. Notwithstanding these concerns, the benefits of HEV239 vaccination in preventing hepatitis E in pregnant women, who face high morbidity and mortality risks from the infection, are substantial, particularly in disadvantaged regions [13].

The findings from these studies highlight that while HEV239 demonstrates high efficacy in preventing HEV infection, the observed association with an increased miscarriage risk when administered around the time of conception necessitates careful consideration in public health policies. Current evidence indicates that HEV239 administration should be avoided during pregnancy or immediately preceding conception unless the pregnancy status can be conclusively determined [13]. This recommendation aligns with the World Health Organization’s Strategic Advisory Group of Experts (SAGE), which advises against routine HEV239 administration in pregnancy until further safety data becomes available [15].

The HEV vaccine has been successfully developed and clinically validated. Expanding vaccination coverage is crucial for reducing infection rates. To align with the global outbreak response strategy, issues such as WHO pre-qualification, age range expansion, emergency immunization schedule optimization, and vaccine transportation and administration must be resolved. Further clinical trials are needed to evaluate the vaccine’s benefits and safety in high-risk groups, including pregnant women, individuals with chronic liver disease, and children under 16. Despite challenges, the successful deployment in South Sudan demonstrates that hepatitis E vaccination is feasible in complex emergencies, potentially accelerating future HEV vaccine use [16]. The advancement of a hepatitis E vaccine necessitates tackling the disease’s complexities, particularly in vulnerable populations such as pregnant and immunocompromised individuals. The Phase 2 trial of Hecolin^®^ in Pakistan is critical for assessing its safety and efficacy in pregnant women, including passive immunity through transplacental transfer. Additional research is imperative to evaluate the vaccine’s effectiveness and safety in those with advanced liver fibrosis or cirrhosis, solid organ transplant recipients, and immunocompromised patients. Longitudinal studies are needed to ascertain the protection longevity and the ideal timing for booster shots. Addressing the unique needs of these groups is vital for reducing hepatitis E’s impact and enhancing outcomes for those who are most at risk [17].

As the only commercially available hepatitis E (HE) vaccine worldwide, Hecolin is a high priority for vulnerable populations, including women of childbearing age. Vaccinating this group is cost-effective from a societal perspective, particularly in epidemic regions [18,19].

Comparing HEV239 with other maternal vaccines, such as influenza and tetanus, diphtheria, and pertussis (Tdap), reveals that vaccination during pregnancy has been widely considered safe and effective. Studies have shown that influenza vaccination during pregnancy reduces maternal morbidity and offers passive immunity to newborns. Similarly, Tdap vaccines have proven safe and effective in preventing pertussis-related complications in neonates. Incorporating lessons from these vaccines can guide HEV239 vaccination strategies to mitigate risks while maximizing benefits [20,21,22].

Hormonal and immunological changes during pregnancy can modulate vaccine responses, potentially explaining the observed risks. For example, evidence from studies on maternal vaccination suggests that immune hypersensitivity could play a role in vaccine-related adverse outcomes [23]. Additionally, placental immune interactions may mediate the miscarriage risks associated with vaccine administration near conception [24].

HEV outbreaks disproportionately affect women in low-resource settings, particularly in regions with poor sanitation and healthcare infrastructure. Successful HEV vaccination campaigns, like those conducted in displaced populations during the South Sudan outbreak, illustrate the potential of vaccination as a public health tool in emergencies [25]. Expanding these efforts to stable but vulnerable populations requires overcoming logistical challenges, as highlighted in studies on vaccine deployment during pandemics.

Ethical considerations also play a pivotal role in vaccine research. The exclusion of pregnant women from clinical trials has historically hindered the development of evidence-based vaccination guidelines. Recent efforts to include pregnant women in vaccine studies underscore the need for ethically inclusive research designs to ensure equitable healthcare access [26]. In endemic regions, malnutrition, and related factors can significantly impact immune responses to vaccines. Studies have shown that micronutrient deficiencies, such as vitamin A and zinc, reduce vaccine efficacy and increase the severity of infectious diseases [27]. Evaluating the role of nutritional supplementation in enhancing HEV vaccine efficacy is an area that warrants further investigation.

This systematic review and meta-analysis mark the first exhaustive synthesis examining the safety and efficacy of the HEV239 hepatitis E vaccine, with a specific focus on women of reproductive age, including potential pregnancies. The study’s strength stems from its meticulous methodology, employing the PRISMA guidelines for data collection and utilizing an extensive array of databases and ‘’Gray’’ literature to establish a thorough evidence foundation. By concentrating on a susceptible demographic, namely women of childbearing age, this analysis offers crucial insights into the vaccine’s impact on maternal and gestational outcomes. The application of Cochrane’s Risk of Bias 2 tool and the Newcastle Ottawa Scale for quality assessment enhances the credibility of the incorporated studies, while the quantitative synthesis via meta-analysis strengthens the statistical validity of findings regarding rates of miscarriage, stillbirth, and elective termination.

Several limitations of this systematic review must be acknowledged. First, prior seropositivity in the trial populations may have influenced the results, as these studies were conducted in HEV-endemic regions where pre-existing immunity is common. This factor could impact vaccine efficacy and safety outcomes, particularly in seropositive individuals. Second, malnutrition, prevalent in many HEV-endemic areas, may have affected immune responses and the severity of HEV disease, yet this was not consistently reported or accounted for in the studies. Third, environmental and sociodemographic factors, such as those observed in the Sudan trial conducted in a refugee camp for internally displaced populations, introduce additional challenges. Refugee settings are characterized by poor sanitation, overcrowding, and limited healthcare access, which may influence HEV transmission dynamics and vaccine performance, limiting the generalizability to stable populations. Moreover, heterogeneity in study designs, population demographics, and follow-up durations reduces the comparability of the findings and introduces uncertainty in the pooled analysis. Potential publication bias, particularly from smaller studies with negative results, cannot be excluded and may have led to overestimating the vaccine’s efficacy and safety. Furthermore, the studies predominantly provided short-term safety and efficacy data, with a dearth of longer-term assessments of HEV239′s impact on neonatal and maternal health outcomes. The review’s language restriction to English and exclusion of non-primary research articles may have constrained its capacity to capture relevant findings from diverse sources.

Future research should prioritize studies investigating the biological mechanisms underlying potential miscarriage risks, with particular emphasis on immune responses and placental impacts of HEV239 during the periconceptional period and early gestation. Longitudinal studies with extended follow-up periods are imperative to assess potential delayed effects on maternal and neonatal health postvaccination. Timing-specific safety trials examining the effects of HEV239 administration at various stages of pregnancy would provide valuable guidance for optimizing vaccination protocols for women of reproductive age. Expanding the scope to include non-English studies and additional ‘‘Gray’’ literature could offer a more comprehensive and globally representative understanding of vaccine safety and efficacy across diverse populations.

## 5. Conclusions

This meta-analysis yields crucial insights into the efficacy and safety profile of the HEV239 hepatitis E vaccine in women of reproductive age, representing a significant advancement in understanding the vaccine’s performance within this high-risk population. The results corroborate the vaccine’s substantial protective effect against hepatitis E, which is consistent with its documented immunogenicity. 

Nevertheless, the analysis also reveals important safety considerations, particularly the increased risk of miscarriage associated with vaccine administration near the time of conception. Although the underlying mechanisms remain to be elucidated, these findings necessitate a cautious approach when administering HEV239 to women contemplating pregnancy. 

The potential benefits of preventing hepatitis E, a major contributor to maternal morbidity and mortality, must be judiciously balanced against the possible pregnancy risks. Further research should prioritize mechanistic investigations and timing-specific safety trials to optimize vaccination strategies and enhance maternal and neonatal health outcomes.

## Figures and Tables

**Figure 1 vaccines-13-00053-f001:**
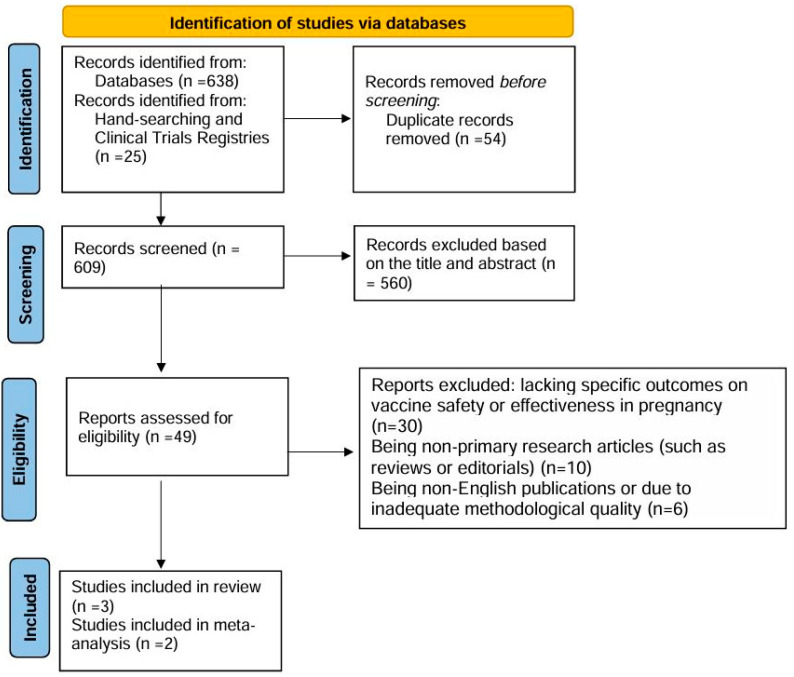
PRISMA.

**Figure 2 vaccines-13-00053-f002:**
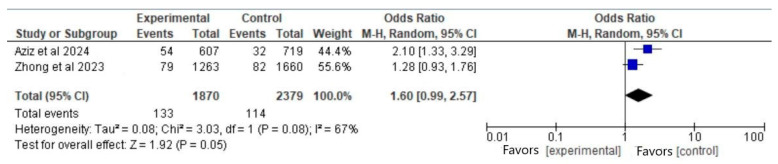
Forest plot for the outcome of miscarriage (vaccinated participants before pregnancy vs. nonvaccinated), based on data from Aziz et al. (2024) [3] and Zhong et al. (2023) [5].

**Figure 3 vaccines-13-00053-f003:**
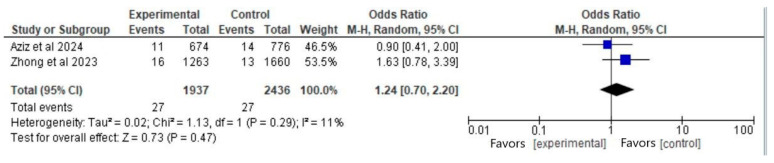
Forest plot for the outcome of stillbirth (vaccinated participants before pregnancy vs. nonvaccinated), based on data from Aziz et al. (2024) [3] and Zhong et al. (2023) [5].

**Figure 4 vaccines-13-00053-f004:**
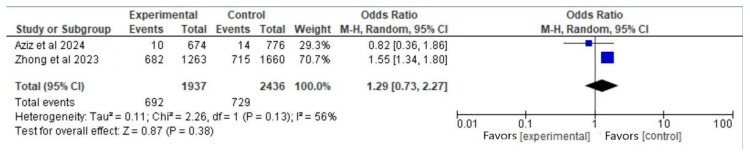
Forest plot for the outcome of elective termination (vaccinated participants before pregnancy vs. nonvaccinated), based on data from Aziz et al. (2024) [3] and Zhong et al. (2023) [5].

**Table 1 vaccines-13-00053-t001:** Summary of the studies’ characteristics.

Author(s)	Year	Location	Study Design	Sample Size	Study Sample	Intervention Details	Primary Outcome	Secondary Outcome	Summary of Results
Aziz et al. [3]	2024	Rural Bangladesh	Double-blind, cluster-randomized trial	19,460	Non-pregnant women aged 16–39	HEV239 vs. Hepa-B vaccine, 3 doses	Miscarriage risk among vaccinated	Fetal loss outcomes (miscarriage, stillbirth, elective termination)	Elevated miscarriage rates were observed in the HEV239 group: miscarriage occurred in 8.9% of pregnancies when vaccinated within 90 days before conception (aRR 1.9, *p* = 0.013) and 10.5% among those who were vaccinated during pregnancy (aRR 2.1, *p* = 0.036). No increased risk was noted for stillbirth or elective termination.
Zhong et al. [5]	2023	China	Phase 3 clinical trial, post hoc analysis	1263 HEV, 1260 HPV	Healthy women aged 18–45	HEV239 vs. HPV (Cecolin), 3 doses	Maternal and fetal safety during pregnancy	Adverse pregnancy outcomes (miscarriage, stillbirth, neonatal abnormalities)	HEV239 did not show an increased risk of abnormal fetal loss compared to the HPV vaccine (odds ratio, 0.80 for fetal loss; 95% CI, 0.38–1.70), and no significant differences in neonatal abnormalities were observed. This suggests no elevated risk of miscarriage or fetal abnormalities when HEV239 is administered inadvertently during pregnancy.
Nesbitt et al. [13]	2024	Bentiu, South Sudan	Observational study	2741	Pregnant women (14–45 years)	Mass vaccination with HEV239	Fetal loss comparison between vaccinated and unvaccinated	Miscarriage and stillbirth risk	Vaccination during pregnancy did not increase cumulative fetal loss risk, with a rate of 7.2% (95% CI 5.6–8.7) among vaccinated women compared to 6.1% (95% CI 3.7–9.2) in unvaccinated women. This study supports HEV239’s safety during pregnancy within a high-risk, outbreak setting.

aRR: adjusted relative risk; CI: confidence interval; HEV: hepatitis E virus; HEV239: hepatitis E vaccine (HEV239); HPV: human papillomavirus.

## Data Availability

The data presented in this study are available within the article.

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
