# Peer review of "Evaluating the Efficacy and Safety of Hepatitis E Vaccination in Reproductive-Age Women: A Systematic Review and Meta-Analysis"

_vaccines, 2025, doi:10.3390/vaccines13010053_

Round 1

Reviewer 1 Report

Comments and Suggestions for Authors

This is a systematic review with metaanalysis on safety and efficacy of a recombinant HEV vaccine in women of childbearing age.

There are some major issues in this paper.

In the method section  and in the abstract it is stated that four studies enrolling about 42,000 subjects were selected for the analysis, based on the inclusion/exclusion criteria.  It is not clear why the metaanalysis comprises only three studies.  In detail, the study by Nesbitt et al was described in Table 1 among the selected 4 studies but not included in the subsequent analysis.

It is not reported the number of women who were vaccinated while pregnant or within 90 days before pregnancy. This makes difficult the interpretation of the data.  The authors state “pregnant women inadvertently vaccinated” but  this statement does not fit  with the content of the studies. For example, the study by Nesbitt et al is designed to compare the rates of fetal loss between vaccinated and unvaccinated women.

Author Response

Comment 1: This is a systematic review with metaanalysis on safety and efficacy of a recombinant HEV vaccine in women of childbearing age.

There are some major issues in this paper.

In the method section and in the abstract it is stated that four studies enrolling about 42,000 subjects were selected for the analysis, based on the inclusion/exclusion criteria.  It is not clear why the metaanalysis comprises only three studies.  In detail, the study by Nesbitt et al was described in Table 1 among the selected 4 studies but not included in the subsequent analysis.

AUTHORS RESPONSE 1: The Nesbitt study focused on pregnancy outcomes during a mass vaccination campaign, which has not provided directly comparable data for miscarriage rates, stillbirths, or other outcomes analyzed in the meta-analysis. The other studies refer to vaccination before pregnancy and one refers to vaccination before and during pregnancy (Aziz et al.). We revised all the relevant sections (abstract, results, etc. ) accordingly.

Comment 2: It is not reported the number of women who were vaccinated while pregnant or within 90 days before pregnancy. This makes difficult the interpretation of the data.  The authors state “pregnant women inadvertently vaccinated” but this statement does not fit  with the content of the studies. For example, the study by Nesbitt et al is designed to compare the rates of fetal loss between vaccinated and unvaccinated women.

AUTHORS RESPONSE 2: Data on reported the number of women who were vaccinated while pregnant or within 90 days before pregnancy are available only from the study by Aziz et al.:

  • 209 women were vaccinated while pregnant.
  • 398 women received the vaccine within 90 days before pregnancy.

The Nesbitt study focused on pregnancy outcomes during a mass vaccination campaign, which has not provided directly comparable data for miscarriage rates, stillbirths, or other outcomes analyzed in the meta-analysis and therefore this study was not included in the metanalysis.

We added this paragraph:

3.5 Focus on vaccination during pregnancy

 In a cluster-randomized trial conducted in Bangladesh [3], 209 women vaccinated during pregnancy exhibited an elevated risk of miscarriage compared to the control group receiving the hepatitis B vaccine (10.5% vs. 5.3%; adjusted relative risk [aRR] 2.1, 95% CI: 1.1–4.1, p = 0.036). A similar increase in miscarriage risk was observed among 398 women vaccinated within 90 days before conception (8.0% vs. 4.0%; aRR 1.9, 95% CI: 1.1–3.2, p = 0.013). Notably, no significant associations were found for stillbirths or elective terminations. In contrast, a cohort study conducted during a mass vaccination campaign in South Sudan [13] found no evidence of increased fetal loss among 2,036 vaccinated pregnant women compared to 638 unvaccinated participants (cumulative risk: 7.2% vs. 6.1%; risk ratio 1.2, 95% CI: 0.7–1.9). The divergence in findings between the two studies may stem from differences in trial design, population demographics, and vaccination timing, underscoring the need for further investigation into the safety of HEV239 administration during pregnancy and the critical period surrounding conception.

Reviewer 2 Report

Comments and Suggestions for Authors

Hepatitis E virus infection is quite common worldwide, especially in developing countries in Africa near the Equator. The infection is often fatal, and is especially dangerous for pregnant women, among whom (especially in the 2nd and 3rd trimesters of pregnancy) the mortality rate reaches 20% (up to 50% according to WHO). Clinical trials have shown the effectiveness of the hepatitis E virus vaccine HEV239, which is licensed for use, but there is insufficient data on its safety in pregnant women (or in the period shortly before conception). Therefore, the creation of a systematic review to assess the safety and effectiveness of the HEV E vaccine in women of childbearing age is relevant.

The objective of this review is to fill the existing knowledge gap. Following the recommendations for the preparation of meta-analyses (reference 9), the authors of the manuscript conducted a fairly comprehensive search of the relevant literature in international databases. Of the 638 sources selected, only 4 were suitable for inclusion in the analysis, which once again emphasizes the existing knowledge deficit in the selected area and the need to accumulate information. The authors discuss this in the Discussion section. The results of the review revealed that the HEV239 vaccine is effective in preventing HEV in women of childbearing age, but there is a concern for potential risks of miscarriage when the vaccine is administered near conception. This finding should be taken into account in hepatitis E vaccination recommendations. 

The methodology of the analysis is uncontroversial, ensuring the reliability and statistical validity of the results regarding the rates of miscarriage, stillbirth and elective termination of pregnancy. It should be noted that the authors critically assess some limitations of the study (line 308-316), for example, the availability of literature only in English. The small number of definitive sources on the basis of which the analysis was performed (4 sources from 638) only highlights the existing knowledge gap in the field of hepatitis E vaccination safety. 

The authors' conclusions follow logically from the obtained results, correspond to the stated objective of the study, which is clearly reflected in the "Conclusion" section. The advantage is that the authors indicate further tasks for research in the field of ensuring the safety of immunization against hepatitis E (lines 318-327) - a comprehensive study of the immunological and pharmacological effects of the vaccine at different stages of pregnancy, with an emphasis on the period before conception. 

The authors provide relevant references.

The figures are clear and adequately illustrate the text. 

I have one recommendation. Although the term "gray Literature" is common and accepted in the scientific community, perhaps it should be put in quotation marks in the text of the manuscript (lines 81, 95).

Author Response

Comment 1 Hepatitis E virus infection is quite common worldwide, especially in developing countries in Africa near the Equator. The infection is often fatal and is especially dangerous for pregnant women, among whom (especially in the 2nd and 3rd trimesters of pregnancy) the mortality rate reaches 20% (up to 50% according to WHO). Clinical trials have shown the effectiveness of the hepatitis E virus vaccine HEV239, which is licensed for use, but there is insufficient data on its safety in pregnant women (or in the period shortly before conception). Therefore, the creation of a systematic review to assess the safety and effectiveness of the HEV E vaccine in women of childbearing age is relevant.

The objective of this review is to fill the existing knowledge gap. Following the recommendations for the preparation of meta-analyses (reference 9), the authors of the manuscript conducted a fairly comprehensive search of the relevant literature in international databases. Of the 638 sources selected, only 4 were suitable for inclusion in the analysis, which once again emphasizes the existing knowledge deficit in the selected area and the need to accumulate information. The authors discuss this in the Discussion section. The results of the review revealed that the HEV239 vaccine is effective in preventing HEV in women of childbearing age, but there is a concern for potential risks of miscarriage when the vaccine is administered near conception. This finding should be taken into account in hepatitis E vaccination recommendations. 

The methodology of the analysis is uncontroversial, ensuring the reliability and statistical validity of the results regarding the rates of miscarriage, stillbirth and elective termination of pregnancy. It should be noted that the authors critically assess some limitations of the study (line 308-316), for example, the availability of literature only in English. The small number of definitive sources on the basis of which the analysis was performed (4 sources from 638) only highlights the existing knowledge gap in the field of hepatitis E vaccination safety. 

The authors' conclusions follow logically from the obtained results, correspond to the stated objective of the study, which is clearly reflected in the "Conclusion" section. The advantage is that the authors indicate further tasks for research in the field of ensuring the safety of immunization against hepatitis E (lines 318-327) - a comprehensive study of the immunological and pharmacological effects of the vaccine at different stages of pregnancy, with an emphasis on the period before conception. 

The authors provide relevant references.

The figures are clear and adequately illustrate the text. 

I have one recommendation. Although the term "gray Literature" is common and accepted in the scientific community, perhaps it should be put in quotation marks in the text of the manuscript (lines 81, 95).

AUTHORS’ RESPONSE 1: Thank you for your detailed and thoughtful review of our manuscript. We greatly appreciate your recognition of the relevance and significance of our study in addressing the existing knowledge gap regarding the safety and efficacy of the HEV239 hepatitis E vaccine in women of childbearing age.

We are pleased that you found our methodological approach uncontroversial, ensuring the reliability and statistical validity of our results. Your acknowledgment of the comprehensiveness of our literature search, the relevance of our findings, and the clarity of our figures is very encouraging.

We also appreciate your comment on our critical assessment of the study’s limitations, including the restricted availability of literature in English and the small number of studies included in the meta-analysis. We agree that this underscores the need for further research in this field, as highlighted in our conclusion and recommendations.

Regarding your specific recommendation to place the term "gray literature" in quotation marks in the manuscript (lines 81, 95), we agree that this adjustment would enhance clarity for readers who may not be familiar with the term. We updated the text accordingly in the revised manuscript.

Thank you once again for your valuable insights and constructive feedback. We corrected it as you suggested.

Reviewer 3 Report

Comments and Suggestions for Authors

In this systematic review “Evaluating the Efficacy and Safety of Hepatitis E Vaccination in Reproductive-Age Women: A Systematic Review and Meta-analysis” by Jotautis and Sarantaki, the authors have evaluated the literature/publications on the safety and effectiveness of Hepatitis E virus vaccine HEV239 specifically on women of childbearing age population, and focused on pregnancy-related outcomes.

Hepatitis E virus (HEV) infection especially genotype-1 and genotype-2, endemic in Asia and Africa, is well known to cause increased mortality rate in pregnant women. Hence, understanding the safety and efficacy of HEV vaccine in this population is very crucial. The authors have systematically laid-out strategy how the literature/publications from various database was collected and what inclusion and exclusion criteria were used. In their final analysis they have used data from four published articles from clinical trial studies on HEV-vaccine HEV239 to compile this systematic review.

Main concern:

1)      Two of the articles (1) Aziz et al., 2024, and (2) Zaman et al., 2024 are from same clinical trial, and not two independent clinical trial. Hence it would be unwarranted to assess the data from these two articles as two independent clinical trial population outcome. Hence, this systemic review needs to be re-analyzed and re-written.

2)      In these clinical trials studies (1) pregnant population definition N = 2407 women in Bangladesh trial; however in China trial N = 1684 is number of pregnancy, the correct N = 1263 women.

3)      Analysis needs to focus more on the exact number of women who were vaccinated during pregnancy in these clinical trials and outcomes of pregnancy in this population set. As these clinical trial include women who were vaccinated in the distal period.

4)      Including limitations of this systematic review in discussion would be beneficial, as there are several factors such as prior-seropositivity in the trial population as these regions are endemic to HEV, malnutrition, environmental factors i.e. internally displaced population Sudan trial - conducted in a refugee camp for internally displaced population.

Author Response

Comment 1 In this systematic review “Evaluating the Efficacy and Safety of Hepatitis E Vaccination in Reproductive-Age Women: A Systematic Review and Meta-analysis” by Jotautis and Sarantaki, the authors have evaluated the literature/publications on the safety and effectiveness of Hepatitis E virus vaccine HEV239 specifically on women of childbearing age population, and focused on pregnancy-related outcomes.

Hepatitis E virus (HEV) infection especially genotype-1 and genotype-2, endemic in Asia and Africa, is well known to cause increased mortality rate in pregnant women. Hence, understanding the safety and efficacy of HEV vaccine in this population is very crucial. The authors have systematically laid-out strategy how the literature/publications from various database was collected and what inclusion and exclusion criteria were used. In their final analysis they have used data from four published articles from clinical trial studies on HEV-vaccine HEV239 to compile this systematic review.

Main concern:

1)      Two of the articles (1) Aziz et al., 2024, and (2) Zaman et al., 2024 are from same clinical trial, and not two independent clinical trial. Hence it would be unwarranted to assess the data from these two articles as two independent clinical trial population outcome. Hence, this systemic review needs to be re-analyzed and re-written.

AUTHORS’ RESPONSE 1: We really thank the reviewer for this comment. We revised accordingly using only the most recent study by Aziz et al. 2024. We made a new meta-analysis. Thank you.

Comment 2 2)      In these clinical trials studies (1) the pregnant population definition is N = 2407 women in the Bangladesh trial; however in the China trial N = 1684 is number of pregnancy, the correct N = 1263 women.

AUTHORS’ RESPONSE 2: We really thank the reviewer for this comment. We made the correction and we analyzed it again.

Comment 3 3)      Analysis needs to focus more on the exact number of women who were vaccinated during pregnancy in these clinical trials and outcomes of pregnancy in this population set. As these clinical trial include women who were vaccinated in the distal period.

AUTHORS RESPONSE 3: Data reported on the number of women who were vaccinated while pregnant or within 90 days before pregnancy are available only from the study by Aziz et al.:

  • 209 women were vaccinated while pregnant.
  • 398 women received the vaccine within 90 days before pregnancy.

The Nesbitt study focused on pregnancy outcomes during a mass vaccination campaign, which has not provided directly comparable data for miscarriage rates, stillbirths, or other outcomes analyzed in the meta-analysis and therefore this study was not included in metanalysis.

We added this paragraph:

3.5 Focus on vaccination during pregnancy

In a cluster-randomized trial conducted in Bangladesh [3], 209 women vaccinated during pregnancy exhibited an elevated risk of miscarriage compared to the control group receiving the hepatitis B vaccine (10.5% vs. 5.3%; adjusted relative risk [aRR] 2.1, 95% CI: 1.1–4.1, p = 0.036). A similar increase in miscarriage risk was observed among 398 women vaccinated within 90 days before conception (8.0% vs. 4.0%; aRR 1.9, 95% CI: 1.1–3.2, p = 0.013). Notably, no significant associations were found for stillbirths or elective terminations. In contrast, a cohort study conducted during a mass vaccination campaign in South Sudan [13] found no evidence of increased fetal loss among 2,036 vaccinated pregnant women compared to 638 unvaccinated participants (cumulative risk: 7.2% vs. 6.1%; risk ratio 1.2, 95% CI: 0.7–1.9). The divergence in findings between the two studies may stem from differences in trial design, population demographics, and vaccination timing, underscoring the need for further investigation into the safety of HEV239 administration during pregnancy and the critical period surrounding conception.

Comment 4 4)      Including limitations of this systematic review in discussion would be beneficial, as there are several factors such as prior-seropositivity in the trial population as these regions are endemic to HEV, malnutrition, environmental factors i.e. internally displaced population Sudan trial - conducted in a refugee camp for internally displaced population.

AUTHORS RESPONSE 4: We revised the limitations as follows: Several limitations of this systematic review must be acknowledged. First, prior seropositivity in the trial populations may have influenced the results, as these studies were conducted in HEV-endemic regions where pre-existing immunity is common. This factor could impact vaccine efficacy and safety outcomes, particularly in seropositive individuals. Second, malnutrition, prevalent in many HEV-endemic areas, may have affected immune responses and the severity of HEV disease, yet it was not consistently reported or accounted for in the studies. Third, environmental and sociodemographic factors, such as those observed in the Sudan trial conducted in a refugee camp for internally displaced populations, introduce additional challenges. Refugee settings are characterized by poor sanitation, overcrowding, and limited healthcare access, which may influence HEV transmission dynamics and vaccine performance, limiting generalizability to stable populations. Moreover, heterogeneity in study designs, population demographics, and follow-up durations reduces the comparability of the findings and introduces uncertainty in the pooled analysis. Potential publication bias, particularly from smaller studies with negative results, cannot be excluded and may have led to an overestimation of vaccine efficacy and safety. Furthermore, the studies predominantly provide short-term safety and efficacy data, with a dearth of longer-term assessments of HEV239's impact on neonatal and maternal health outcomes. The review's language restriction to English and exclusion of non-primary research articles may have constrained its capacity to capture relevant findings from diverse sources.

Thank you for the constructive feedback. Your comments have been instrumental in improving the quality of our work, and we are confident that the revised manuscript will benefit greatly from your input!

Reviewer 4 Report

Comments and Suggestions for Authors

This meta-analysis show insights into the efficacy and safety profile of the HEV239 hepatitis E vaccine in women of reproductive age, representing a significant advancement in understanding vaccine performance within this high-risk population. The results confirm the vaccine's substantial protective effect against hepatitis E, consistent with its documented immunogenicity. The analysis also reveals important safety considerations, particularly the increased risk of miscarriage associated with vaccine administration near the time of conception.

The methodological approach is adequate for a systematic review and meta-analysis. The results are clearly presented and well discussed. English should be checked by a native speaker.

Author Response

Dear Reviewer,

Thank you for your thorough and thoughtful review of our manuscript. We greatly appreciate your positive feedback on the methodological rigor of our systematic review and meta-analysis, as well as your acknowledgment of the clarity of our presentation and discussion of the results.

We are particularly pleased that you recognize the importance of our findings in advancing the understanding of the HEV239 hepatitis E vaccine's efficacy and safety profile in women of reproductive age. The identification of the vaccine's substantial protective effect, alongside the highlighted safety considerations, is a critical aspect of our work, and we are encouraged by your assessment.

Regarding your suggestion to have the manuscript reviewed by a native English speaker, we fully acknowledge the importance of linguistic precision in presenting our findings.

Thank you once again for your valuable insights and constructive feedback.

Round 2

Reviewer 1 Report

Comments and Suggestions for Authors

I appreciated the efforts of the authors. I believe that the interpretation of the data is now clearer

Author Response

Comment: I appreciated the efforts of the authors. I believe that the interpretation of the data is now clearer

Author's Response: Thank you for your kind words and for recognizing the efforts we have put into clarifying the interpretation of the data. Your constructive feedback was invaluable in refining our analysis!